# Engineered Healing: Synergistic Use of Schwann Cells and Biomaterials for Spinal Cord Regeneration

**DOI:** 10.3390/ijms26167922

**Published:** 2025-08-16

**Authors:** Theo Andriot, Mousumi Ghosh, Damien D. Pearse

**Affiliations:** 1The Miami Project to Cure Paralysis and Department of Neurological Surgery, Miller School of Medicine, University of Miami, Miami, FL 33136, USA; txa825@med.miami.edu (T.A.); mghosh@med.miami.edu (M.G.); 2The Neuroscience Program, Miller School of Medicine, University of Miami, Miami, FL 33136, USA; 3The Interdisciplinary Stem Cell Institute, Miller School of Medicine, University of Miami, Miami, FL 33136, USA

**Keywords:** spinal cord injury (SCI), cell therapy, Schwann cells, biomaterials, regeneration, cell survival

## Abstract

Spinal cord injury (SCI) remains a devastating neurological condition characterized by loss of sensory, motor and autonomic function. Despite decades of research, no FDA-approved regenerative therapies currently exist to restore lost function following SCI. Schwann cells (SCs) support axon regeneration, remyelination, and neuroprotection after SCI, with their therapeutic potential validated in clinical trials demonstrating safe and feasible transplantation in humans. Although SC transplantation has shown promising results, challenges remain, including modest graft survival, limited host integration, and restricted migration that collectively contribute to constrain efficacy. To address these limitations, biomaterial scaffolds have been explored as synergistic platforms to enhance SC delivery and function. When combined with natural or synthetic biomaterials such as hydrogels, nanofiber scaffolds, or ECM-mimetic matrices, SCs demonstrate improved survival, retention, spatial distribution, and regenerative activity. The intrinsic regenerative properties of SCs, first demonstrated in models of peripheral nerve injury, make them particularly well-suited for neural repair of the central nervous system (CNS) compared to other cell types and their effectiveness can be enhanced synergistically when combined with biomaterials. These constructs not only provide structural support but also modulate the lesion microenvironment, enhance axon growth and improve SC integration with host tissue. Combinatorial approaches incorporating biomaterials with SCs are emerging as next-generation strategies to optimize repair for clinical translation. This review focuses on current progress in SC-based therapies combined with biomaterials, highlighting key preclinical advances, clinical translation efforts, and the path forward toward effective regenerative interventions for SCI.

## 1. Introduction

Injuries to the adult mammalian central nervous system (CNS), such as spinal cord injury (SCI), are complex and devastating conditions that lead to extensive neurological damage and long-term sensory and motor disabilities. Data from the World Health Organization (WHO) indicates that 20.6 million individuals presently live with SCI, with an incidence of 0.9 million new SCI per year worldwide [1]. Besides being a major cause of long-term disability and premature death, SCI causes a substantial social and economic toll. The US National SCI Statistical Center estimated lifelong costs for SCI vary from US $1 to 5 million per person, depending on the location and severity of the lesion [2]. Major causes of traumatic SCI include car accidents, falls and violence [3,4].

Understanding and treating SCI remains inherently challenging due to the heterogeneity of lesion morphologies [5]. The severity, characteristics, and potential for recovery are shaped by a dynamic interplay of cellular and molecular events occurring within the distinct compartments of the lesion [6,7,8].

SCI pathophysiology is classically divided into two sequential phases [9]: the primary injury and the secondary injury. The primary injury is confined to the site of trauma and results from mechanical forces such as rupture, contusion, or compression, leading to immediate destruction of spinal tissue along with resulting disruption of the vascular system to produce hemorrhage and ischemia. This initial mechanical insult initiates a complex cascade of cellular, molecular, and biochemical responses. Based on the foundational work by Allen [10], these processes trigger a secondary injury marked by the progressive degeneration of tissue around the lesion and worsening of neurological deficits. During this secondary phase, the breakdown of the blood–spinal cord barrier (BSCB) triggers extensive vascular and inflammatory responses, further exacerbating tissue damage [11]. These include extended hemorrhage, immune cell infiltration, oxidative stress, excitotoxicity, and glial activation, all contributing to lesion expansion and impaired recovery. This biochemical milieu induces axon demyelination and neuronal death, which in turn releases growth-inhibitory molecules into the lesion environment [12,13]. Concurrently, the inflammatory response leads to the formation of a dense glial scar enriched with extracellular matrix (ECM) inhibitors, such as chondroitin sulfate proteoglycans (CSPG) [6,14,15,16]. At the lesion core, a fluid-filled cystic cavity often develops. In contrast to the peripheral nervous system (PNS), which retains robust regenerative capacity [17], CNS axons have severely limited ability to regenerate into this complex lesion milieu, and neuronal loss is not compensated by cell replacement. As a result of this constrained regenerative environment, endogenous functional recovery after traumatic SCI is typically unsuccessful.

Following SCI, re-establishing functional neural circuits remains profoundly challenging, and to date no restorative therapeutics have proven reliably effective in clinical trials for receiving FDA approval [18,19,20]. Due to the lack of approved regenerative therapies, research is increasingly turning to innovative strategies such as tissue engineering, and biomaterials.

This review focuses specifically on the convergence of biological and engineering strategies through the combined application of SC transplantation with biomaterial scaffolds. SCs, the myelinating glia of the PNS, are well-established for their capacity to support axonal regrowth and remyelination in the injured nervous system, and SC transplantation has shown promising outcomes in both PNS and CNS lesions, from preclinical rodent models of SCI to early-stage clinical trials [21,22,23,24]. When delivered alongside engineered scaffolds, such as hydrogels, aligned nanofibers, and ECM–mimicking materials, they create a permissive microenvironment that enhances cell survival, directs axonal regrowth, promotes remyelination, and mitigates post-injury inhibitory signaling [25,26,27] (Figure 1). Preclinical studies have demonstrated that SC-seeded biomaterial scaffolds can reduce cystic cavitation, increase axonal penetration, and improve functional outcomes [28] and these will be further elaborated upon in this review. Additionally, recent advances in scaffold design, including self-assembling peptide nanofibers, laser-aligned fibers, and hyaluronic acid-based hydrogels, have been shown to provide biochemical and topographical cues to mimic the native ECM to facilitate SC alignment and enhanced neurite outgrowth [28]. Bioengineered materials for co-implantation with SCs are also manufactured to improve support for vascularization, synaptic integration, and sustained bioactivity, critical for long-term repair. Finally, although we will discuss the encouraging results of the early preclinical studies exploring the combined use of SCs and biomaterials, we will also review the translational challenges that remain. These include the scalability of autologous SC isolation, immunogenicity of scaffold materials, integration of these materials with host tissue, and the need for robust and reproducible functional outcomes in diverse patient populations [29,30].

## 2. The Regenerative Role of SCs in the Peripheral Nervous System and Their Adaption as a Reparative Cell Intervention for the Injured Spinal Cord

Cell-based therapies hold strong potential for repairing traumatic SCI, and SCs are of particular interest due to their critical role in promoting axonal regeneration, remyelination, and neuroprotection. In the PNS, regeneration is governed by a coordinated series of events involving both injured neurons and non-neuronal cells, especially SCs [31]. Upon axon injury, the calcium influx activates proteases that reorganize the cytoskeleton and fragment axons. SCs then rapidly dedifferentiate into a repair phenotype, demyelinate injured fibers, proliferate and recruit macrophages to clear debris and release cytokines that promote SC and fibroblast proliferation. These SCs migrate and align their cell processes to form specialized, interdigitating columns, called the Bands of Büngner, to guide regenerating axons [32] while secreting neurotrophic factors such as brain-derived neurotrophic factor (BDNF), glial cell line-derived neurotrophic factor (GDNF), and nerve growth factor (NGF) that support axon growth and remyelination [33]. The ECM is concurrently remodeled marked by elevated laminin expression, enhancing axonal elongation, SC migration, and myelination. This regenerative cascade is complemented by intrinsic neuronal responses, where injury-induced signals from the lesion propagate to the soma, initiating action potentials and a calcium influx that activates signaling cascades and transcription factors such as c-Jun, ATF3, CREB, and STATs [34,35]. These transcriptional programs reinitiate growth-associated genes, adhesion molecules, and cytoskeletal regulators necessary for axonal regrowth. While regeneration in the adult PNS is often incomplete, the role of SCs is a well characterized and critical component of regeneration after injury.

Given their intrinsic regenerative capacities and the ability of SCs to support PNS axon regeneration, and to test the hypothesis that CNS axons could regenerate if provided a permissive environment, seminal work using peripheral nerve implantation into the injured CNS demonstrated that this concept was true [36,37]. To further enhance the axon growth-promoting and guidance properties of transplanted peripheral nerve tissue, subsequent studies have employed a variety of adjunct approaches that have been extensively reviewed [38], including combinatorial approaches involving other cell types such as olfactory ensheathing cells (OECs) through external approaches such as physiotherapy [39,40] to further improve repair and recovery. Recent work shows that SCs formed into 3D spheroids exhibit an enhanced repair phenotype and greater therapeutic efficacy in promoting PNS regeneration compared to dissociated cells [41], harkening back to the benefit provided by whole tissue implants of peripheral nerve. These regenerative attributes have motivated their adaptation for CNS repair, particularly in SCI, where endogenous regenerative capacity is limited. SCs may facilitate new tissue formation, bridge lesion cavities, and exert trophic and immunomodulatory effects that promote axonal plasticity (Table 1). They naturally infiltrate SCI lesions as part of the post-injury response termed “schwannosis” [42], a benign proliferation of SCs within the CNS, and such findings have led to the hypothesis that the transplantation of SCs into the injured spinal cord could aid in repair [43].

Indeed, transplanted SCs have been shown to survive, partially integrate, and myelinate host-derived axons, while modulating inflammation and attracting regenerating fibers. Early preclinical studies using autologous SCs from peripheral nerves demonstrated increased axonal density and modest functional recovery [44,45]. Pearse et al. [21] showed that elevating intracellular cAMP in combination with SC transplantation enhanced axonal growth and significantly improved locomotor outcomes, highlighting the importance of intracellular signaling in overcoming CNS inhibition. The beneficial interplay between cAMP and SCs was also confirmed by other teams [46,47]. In a follow-up study, Pearse et al. [22] reported that SCs transplanted into contused spinal cords survived, migrated within the lesion, and associated with regenerating axons, and these findings were observed in similar investigations [48]. Co-transplantation with olfactory ensheathing glia (OEG) has yielded even better axonal regeneration and function [22,49,50]. Subsequently, Fouad et al. [51] demonstrated that combining SCs and Matrigel with chondroitinase ABC, which degrades inhibitory ECM components, enhances SC migration, axonal growth, and functional outcomes, underscoring the value of combinatorial approaches. Similarly, Williams et al. [52] demonstrated that autologous SC grafts in fluid Matrigel form permissive interfaces across lesion sites that enable axonal crossing and are associated with improved hindlimb motor function, highlighting the role of SCs in establishing scaffold-like structures that promote regeneration.

Despite these advances, limitations persist (Table 1). After SCI, the hostile environment, including glial scarring, inflammation, and inhibitory ECM, impairs SC survival and limits their migration and integration, often confining them to the lesion core. Moreover, harvesting autologous SCs requires invasive biopsies with modest yields, while allogeneic sources carry potential immunogenic risks. Stem-cell–derived SC-like cells from iPSCs or MSCs are under investigation but remain unproven for clinical use. These challenges have led to increasing interest in combining SCs with biomaterial scaffolds to improve cell survival, guide axon growth, and enhance functional integration.

**Table 1 ijms-26-07922-t001:** Benefits and limitations of SC transplantation as a cell-therapeutic for repair of the injured spinal cord.

Therapeutic Advantages	Limitations
Promotes axonal growth [21,53,54]	Modest graft survival and retention [22,55,56]
Enhances remyelination in rodent SCI models [54,57,58]	Limited host integration [29]
Secretes neurotrophic factors [59,60,61]	Senescence, limited expansion in culture [29]
Produces supportive ECM [62]	Produce inhibitory ECM components [63,64]
Exhibits anti-inflammatory properties [65,66]	Poor migration post-implantation [53]
Safe and promising in autologous clinical use [24] Found endogenously within the spinal cord lesion after SCI [67]	Autologous cell availability constraints [68]

## 3. Enhancing the Effectiveness of SC Transplantation After SCI Using Biomaterials

SC transplantation has shown promise in promoting CNS regeneration [22,52,66,69,70,71,72,73,74]; however, efficacy is often limited by substantial challenges associated with cell delivery, host integration and persistence. A major limitation is the significant loss of transplanted cells during or shortly after injection, which results from mitogen withdrawal, mechanical shear stress, cell membrane rupture, anoikis and exposure to the hostile post-injury environment, triggering cell apoptosis or necrosis. These combined factors drastically impair the survival and therapeutic impact of transplanted cells. Combining SCs with biomaterial scaffolds has emerged as a logical and synergistic strategy to address these challenges. Biomaterials offer structural and biochemical support, protect cells during delivery, improve retention and spatial distribution, and promote integration with the host tissue. They can also serve as delivery vehicles for growth factors or drugs, to further provide cytoprotection and to help recreate a permissive microenvironment for regeneration. A consistent finding is that combining SCs with biomaterials yields better outcomes than either approach alone [27]. These improvements include increased axon density and orientation (especially with aligned scaffolds), enhanced remyelination, greater cell survival (notably when cells are encapsulated or embedded), and superior functional recovery in preclinical models [26,75,76]. A study by Olson et al. [77] comparing the efficacy of SCs and NSCs delivered through biomaterial channels in spinal cord transection found that while no significant differences were observed in axon density per channel at an individual level, and no meaningful improvements in functional recovery were observed in their paradigm among cell types, SC-seeded scaffolds trended to greater axonal regrowth than NSC-seeded scaffolds. Moreover, systematic review suggests that different cell types (SCs, OECs, MSCs, NSCs), when combined with suitable biomaterials, can promote functional recovery, but the extent of recovery and mechanisms of repair may vary depending on the cell type used [78,79,80].

The integration of cells with engineered matrices has become a cornerstone strategy in neural tissue engineering. Biomaterials protect transplanted cells from mechanical and biochemical stress, facilitate anchorage, and create conditions that promote regenerative signaling. Foundational work by King et al. [81] on fibronectin-based scaffolds and more recent studies [82,83] confirm that the composition and design of biomaterials influence lesion permissiveness, cell integration, and functional outcome. Biomaterials can be formulated as injectable gels or preformed 3D scaffolds [84,85], tailored in terms of stiffness, degradation kinetics, and bioactivity to match the regenerative needs of injured spinal cord tissue. However, these biomaterials also need to be tailored in terms of porosity, permeability, mesh size, mechanical and rheological properties to support proper molecule interchange between host and graft as well as normal cell function. In standard rheological approaches, the motion of embedded particles within a material is sensitive to their interactions with the surrounding environment. The size and surface chemistry of these particles significantly influence such interactions and therefore impact the experimental outcomes [86]. Key rheological parameters include elastic modulus (e.g., Young’s modulus), storage modulus (G′) and the viscosity of a biomaterial depends on intrinsic factors such as the polymer mesh spacing of the material and external factors like temperature [86,87,88]. Viscosity specifically affects the capacity of the hydrogel to be injected through a needle and diffuse prior to gelation. In the case of Matrigel, rheological studies have shown that the material remains in a liquid state below 6 °C [87], which facilitates pipetting and handling before gelation occurs at physiological temperatures. Considering these characteristics, hydrogels employed in SCI approaches ideally should mimic the mechanical properties of the host tissue. With the stiffness of the spinal cord ranging from 3 to 300 kPa, these hydrogels should be stiff enough to self-assemble, yet soft enough to support cell adhesion, growth, and differentiation [89]. Soft hydrogels (<1 kPa) with low viscosity have been shown to be well suited to promoting tissue regeneration after implantation [89].

Finally, structural features of the biomaterial, such as channels, nanotopography, or capillary architecture, can support directional axonal regrowth, vascularization, and integration [81,82,83,90,91]. Despite their advantages, biomaterials alone are typically insufficient to induce meaningful regeneration [92,93,94,95]. Without active biological or pharmaceutical components, they may act primarily as passive fillers and fail to overcome inhibitory cues or glial scarring. Moreover, inflammation and mechanical mismatch at the implant site may limit long-term function of implanted biomaterials. Even though 3D porous scaffolds and hydrogels can be adapted to irregular or complex lesion sites [96], and biomaterials may serve as platforms for anti-inflammatory drug delivery, concerns remain regarding their intrinsic immunogenicity. Depending on their composition, biocompatibility, and manufacturing processes, some biomaterials can exacerbate inflammation [97] by activating NF-κB signaling and pro-inflammatory cytokine pathways [98], or by inducing sterile inflammation [99]. For this reason, combining biomaterials with cells has been shown to significantly enhance regenerative outcomes by modulating the injury environment and providing sustained biochemical support. A wide range of natural, synthetic, and hybrid biomaterials have been explored as delivery platforms for SCs in SCI models (Table 2), and numerous preclinical studies support this concept. These materials offer several advantages depending on their configuration (e.g., hydrogel vs. scaffold), including structural support, protection of cells from mechanical stress during injection, and the creation of a permissive microenvironment that promotes cell survival, integration, cell–scaffold interactions and axonal regrowth. However, certain limitations must be considered for transplantation, such as challenges in achieving controlled architecture, ensuring long-term mechanical stability, and mitigating potential inflammatory responses [96,100]. Furthermore, these materials often present obstacles to adequate vascularization, which remains a critical hurdle for tissue integration and sustained function. In the ensuing sections we will discuss the different types of biomaterials, natural, synthetic, and hybrid, that have been used in combination with SCs for SCI repair.

**Natural biomaterials** are derived from ECM components and are typically biocompatible, biodegradable, and bioactive. Various studies have demonstrated their ability to support SC viability, differentiation, and neuroregenerative potential. For example, gelatin hydrogels [90] in rat transection models, alginate-based scaffolds combined with SC transplantation and neurotrophic factor delivery [91,101,102], and fibrin matrices ([81], in hemisection models) have shown promising outcomes. These outcomes include improved cell integration and host–graft interface formation, increased neurotransmitter expression, and greater axon growth (including 5-HT axons from raphespinal neurons), that are associated with improved functional recovery, especially when trophic factors are additionally applied [91]. While CST axon regrowth remains challenging with gelatin-based scaffolds [90], fibronectin mats support the development of non-neural elements, such as vascularization [81]. In vitro studies have also highlighted the benefits of combining SCs with fibrin [103] and alginate composites [91,103], to increase cell viability and guide both supraspinal and propriospinal axons through and beyond the lesion site. Zhang et al. [104] developed a HA-chitosan composite hydrogel that enabled sustained release of the neurotrophic factor NGF alongside SCs, leading to robust axonal regrowth, enhanced remyelination, and significant functional recovery.

Several studies have reported that alginate-based materials can enhance neuronal survival and axonal regeneration. Novikov et al. [105] demonstrated that polyhydroxybutyrate (PHB) fibers loaded with alginate and containing SCs in suspension, supported 75% survival of rubrospinal neurons at 8 weeks after cervical hemisection, a significantly higher rate compared to SCI alone or when loaded with Gelfoam as a control. As a natural biomaterial, alginate hydrogels also show promise for axonal regeneration, particularly in capillary hydrogels. Günther et al. [106] reported that bone marrow stromal cells (BMSCs) encapsulated in alginate hydrogels loaded with BDNF resulted in a 17-fold increase in axon density per channel compared to non-loaded controls, accompanied by endogenous SCs that migrated inside the scaffold. These results have confirmed alginate as an interesting biomaterial to combine with SCs. Similar to these BMSC studies, Liu et al. [91] observed a 1.5- to 2-fold increase in axons per channel when SCs were embedded in BDNF-loaded alginate hydrogels compared to alginate alone. They also reported enhanced axonal penetration and long-distance growth, with up to 50% of raphespinal and 80% of propriospinal axons present in the rostral half of the biomaterial. Descending propriospinal axons extended up to 1200 µm beyond the caudal interface of the hydrogel. However, Estrada et al. [107] noted that polyethylene glycol (PEG)-based hydrogels led to even greater axonal growth, with a 6- to 7-fold increase compared to alginate-based hydrogels.

Building on this foundation, recent work has focused on optimizing natural scaffolds to improve therapeutic efficacy. Decellularized peripheral nerve (PN) matrix and injectable PN hydrogels, both in vitro and in vivo rat cervical contusion models, offer tissue-specific biochemical cues that enhance SC survival and regeneration, reduce lesion size, and improve functional recovery [108,109,110]. Agarwal et al. [110] showed that an injectable decellularized PN hydrogel combined with transplanted SCs, enhanced SC survival, increased axonal extension, reduced lesion volume, and improved motor function in rats. The preserved tissue-specific ECM within the matrix was critical for directing SC alignment and interaction with host axons. Nonetheless, these materials remain complex to manufacture and can exhibit lot-to-lot variability. Their compatibility with human use may also be an issue [111].

Chitosan-based scaffolds hold promise for SCI repair due to their biocompatibility, tunable properties, and capacity for chemical modification to enhance mechanical stability and drug delivery [112,113]. Recent studies have highlighted their ability to guide SC alignment and polarization toward a repair phenotype, promoting survival, remyelination, and functional recovery in both in vitro and in vivo models of SCI [114,115]. Cong et al. [115] demonstrated that microgrooved chitosan scaffolds, seeded with SCs derived from skin precursors, improved cell orientation and produced an upregulation of neurotrophic and repair gene expression. Chitosan can be further enhanced by incorporating growth factors; for example, Zeng et al. [116] embedded NGF-loaded microspheres within a chitosan matrix to provide sustained trophic support. However, some chitosan formulations degrade rapidly and may not significantly affect CSPG expression.

Together, these studies highlight the synergistic potential of combining natural ECM-mimicking scaffolds and SCs. They underscore the potential of natural biomaterial-based scaffolds to provide instructive and supportive environments that optimize SC-mediated regeneration in the injured spinal cord.

**Synthetic biomaterials** offer tunable mechanical and chemical properties, making them attractive for supporting SC transplantation in SCI. Polyhydroxybutyrate (PHB) substrates have demonstrated compatibility with SCs, enhancing adhesion and proliferation, up to a 315% increase with fibronectin-treated PHB, and promoting axonal guidance and rubrospinal neuron regeneration in rat cervical hemisection models [105,117].

In thoracic transection models, Oligo[poly(ethylene glycol) fumarate] (OPF^+^) scaffolds with longitudinal channels promoted axonal alignment, sustained axonal regeneration, and myelination, alongside improved locomotor function [26,118]. Siddiqui et al. [25,118] further demonstrated that neurotrophin-loaded OPF^+^ hydrogels enhanced SC alignment and axon regeneration in chronic SCI. Similarly, Chen et al. [119] used GDNF-secreting SCs delivered via OPF^+^ or PEG hydrogels in a thoracic transection model, resulting in enhanced axon growth, remyelination, and motor improvement within three weeks post-injury. However, given that functional improvement was observed as early as 2 weeks post-injury, prior to when substantial axonal regeneration across the biomaterial would have occurred, it is unclear whether the rapid motor recovery resulted from GDNF-mediated neuroprotection or the plasticity of spared spinal circuits to improve outcomes, rather than from true regenerative effects.

Despite these advances, SC migration and integration remain limited. Hejčl et al. [92] showed that positively charged 2-hydroxyethyl methacrylate (HEMA) scaffolds improved tissue integration and axonal regrowth when implanted acutely or in delayed fashion, although no behavioral improvements were reported. PEG-based hydrogels also promote SC-mediated regeneration by mimicking native neural stiffness and reducing inflammation [107]. Nanostructured materials such as graphene oxide have also been shown to enhance SC adhesion and neurite outgrowth in vitro [120].

Marquardt et al. [76] developed an injectable, self-healing, thermo-responsive peptide-based hydrogel (“SHIELD”), characterized by its shear-thinning behavior, allowing the material to flow as a liquid due to the reversible disruption of peptide–peptide interactions, and by rapid self-healing upon cessation of shear as these interactions quickly reform. Use of SHIELD significantly reduced SC loss post-transplantation, achieving up to a 740% increase in viable local cell delivery in a rat cervical contusion model. Similarly, PLGA scaffolds have supported SC survival and alignment [121], and PCL scaffolds with aligned nanofibers have shown promise in promoting axonal entry, SC–astrocyte intermingling, and neurite extension, although functional recovery remains to be demonstrated [122]. Lifka et al. [28] further highlighted that laser-induced surface ripples and aligned nanofibers can direct SC orientation in vitro, informing scaffold design.

Together, these investigations demonstrate how engineered synthetic materials through injectable formats, neurotrophic delivery, gene-modification and nanotopographical cues can synergize with SC transplantation to enhance survival, alignment, axonal regeneration, remyelination, and potentially functional recovery after SCI.

**Hybrid or Functionalized Biomaterials** combine the advantages of both natural and synthetic systems. Gelatin methacrylate (GelMA) hydrogels, which are photo-crosslinkable and derived from gelatin, have been extensively used to protect SCs [123,124,125]. When loaded with SCs derived from pre-degenerated sciatic nerve in spinal cord hemi-transection models, these hydrogels maintained 85% cell viability, significantly improved locomotor recovery and reduced lesion size, partly by inhibiting apoptosis via the p38 MAPK pathway [75].

A study using BD PuraMatrix™, a self-assembling peptide hydrogel, demonstrated that combining the scaffold with human fetal SCs increased cell survival by 25% compared to SCs alone, and enhanced motor function in rats, supporting the importance of scaffold–cell synergy in functional recovery after thoracic contusion [126]. Matrigel, a basement membrane preparation, has shown efficacy in both in vitro and in vivo transection and contusion models by providing a highly supportive environment for SCs and improving graft–host integration, with a reported 10-fold increase in GFAP^+^ processes within the SC bridge [25,52,55,108]. However, both PuraMatrix and Matrigel have limitations due to poorly defined composition, and the murine sarcoma origin of Matrigel restricts clinical translation.

To overcome these limitations, Deng et al. [127] utilized laminin-coated guidance channels filled with SCs and neurotrophic factors, which promoted unidirectional axonal growth in a rat hemisection model, emphasizing the importance of scaffold architecture and biochemical cues. Similarly, Li et al. [128] showed that a modified fibroblast growth factor (FGF)-loaded hydrogel promoted axonal regeneration and remyelination through enhanced SC function in rats after spinal cord transection, resulting in increased BBB scores, although issues with FGF stability and diffusion remain.

Advanced systems such as self-healing hydrogels with controlled drug release [114], and hydrogels incorporating neurotrophic factors (e.g., NT3, BDNF, GDNF) or gene therapies [91,129,130], further illustrate the potential of bioengineered materials to optimize SC function. These studies report up to 84% MBP^+^ coverage within the channel bridge, suggesting substantial myelination. While the use of lentiviral vectors for NT3 or BDNF delivery achieved expression levels within the bridge that were 1000-fold greater than those obtained with non-viral vector delivery [130], axon regeneration in the distal spinal cord remains limited in several models. Similarly, Luo et al. [114] showed that a hybrid self-healing chitosan hydrogel supported remyelination and improved motor function after SCI in vivo. Interestingly, this study also reported an increase in motor evoked potentials (MEPs) amplitude and a reduction in the latency. As MEPs relies on conduction across synapses, the increased electrophysiological properties suggested enhanced neural conduction through the injury site, supporting motor function recovery.

**Summary.** Hybrid and functionalized biomaterials have emerged as promising platforms for SC transplantation in SCI, offering a balance between biological support and engineered precision. Among these, GelMA hydrogels have demonstrated consistently strong performance due to their tunable mechanical properties and inherent bioactivity. When loaded with SCs, GelMA scaffolds have been shown to improve motor recovery, reduce lesion volume, and suppress apoptosis. More recent iterations of GelMA, incorporating conductive or antioxidant components, have further enhanced SC viability, axonal guidance, and electrophysiological function in preclinical models. Injectable hydrogels derived from decellularized porcine PN tissue have also shown high translational potential. These hydrogels, once decellularized and neutralized to remove cellular components and to significantly attenuate immunogenicity, retain native ECM and mimic the mechanical environment of the spinal cord. Hydrogels from decellularized tissue also offer suitable histocompatibility, low immunogenicity, and nontoxic degradation products, all while supporting SC survival and resisting astrocyte-mediated inhibition [100,110].

Similarly, OPF^+^ scaffolds with aligned microchannels have supported directional SC migration and axon regrowth, particularly when enriched with neurotrophic factors such as BDNF or GDNF. These scaffolds are injectable, modular, and compatible with scalable manufacturing processes. HA-based hydrogels, particularly when functionalized with bioactive molecules like laminin or bFGF, have demonstrated the ability to modulate inflammation, reduce glial scar formation, and promote remyelination through enhanced SC integration. Self-healing hydrogel systems have added further value by offering injectable delivery and sustained release of anti-inflammatory or neuroprotective compounds such as curcumin, creating a protective environment for transplanted cells [114,116].

While most studies discussed here focus primarily on histological and behavioral outcomes, electrophysiological properties of the construct are often overlooked, despite their relevance for functional integration. Indeed, electrical conductivity can directly influence key cellular activities such as axonal growth and guidance. Moreover, SC-mediated axon remyelination aims to restore conduction activity. Although underexplored, some biomaterials, such as decellularized biohybrid nerve grafts, have been engineered to increase conductive properties [131]. Combined with SCs, these biohybrid nerve grafts have demonstrated enhanced conductivity and promoted PN regeneration. Future studies would benefit from systematically assessing and integrating electrophysiological readouts, such as conductivity and MEP amplitude, as indicator of capacity of electrical signal conduction to detect SCI and better predict clinical efficacy of combined SC and biomaterial approaches or employ electroconductive materials or electrical stimulation.

Together, these biomaterial systems, particularly GelMA, decellularized nerve ECM, OPF^+^, and functionalized HA, have consistently demonstrated improvements in the effectiveness of SC transplantation. Their injectability, CNS-matched stiffness and biological compatibility with multiple delivery routes, and suitability for clinical manufacturing make them strong candidates for future translation in SC-based therapies for SCI.

**Table 2 ijms-26-07922-t002:** Advantages of combining biomaterials with SC implantation for SCI repair.

	Biomaterial	Advantage	Refs
Natural	Gelatin hydrogel	Supports SC viability (93.2% survival rate at 7 days post encapsulation) and differentiation, reduces inflammation, promotes 5-HT axon regeneration, synaptic contacts, nerve-regeneration-related and growth factor expression, along with motor recovery when combined with trophic factors (BBB = 8 at 8 weeks, while 2 in SCI only).	[90,132,133]
	Alginate-based scaffold+SC+neurotrophic factors	Up to 1.5–2-fold increase in axon number and improved SC survival, allow axonal infiltration extending up to 1200 µm beyond the caudal interface increased BBB score (SCI = 3.33 vs. SC-hydrogel group = 10.89, at 21 days).	[91,101]
	Fibrin and collagen matrices	Biocompatibility, support for axonal regeneration (10% axon growth in fibronectin vs. 7% in fibrin, increased axon length of 1 mm inside fibrin sealant filled-cavity) and SC viability, fast degradation, BBB of 20 at 3 weeks, versus the control BBB of 17–18.	[103,134,135]
	HA-laminin composite hydrogel (NGF or GDNF-loaded)	Sustained neurotrophins release, robust axonal regrowth, enhanced remyelination, functional recovery.	[136]
	Decellularized PN matrix/Injectable PN hydrogel	95% SC viability and increased graft volume, 212% increase in number of axons in the graft, reduced lesion volume and inflammation, BBB score of 11 at 9 weeks in both PN matrix and SC/Matrigel controls.	[108,109,110]
	Chitosan scaffold (micropatterned or NGF-loaded)	SC polarization, survival, proliferation, migration and alignment, increased neurotrophic and repair associated gene expression.	[114,115]
Synthetic	PHB scaffold (fibronectin-coated)	Increased proliferation with 315% increase in SCs number, enhanced axon guidance with 1500 μm growth in conduits, 75% survival of rubrospinal neurons, compared to controls.	[105,117]
	OPF^+^ hydrogel and PEG scaffold with neurotrophins	Increase axon and blood vessel numbers, increased myelination (up to 33% of total axons in channels), BBB score of 3.67 at 4 weeks versus 2.22 without neurotrophins.	[25,26,118,119]
	HEMA scaffold	Improved axonal regrowth, reduced cavity volume (~25 mm^2^ in control to ~5 mm^2^), no behavioral improvement.	[92]
	PEG-based hydrogels	6–7-fold higher axons density, promotes axon regeneration (2.5% Neurofilament intensity in PEG compared to 0.5% in alginate) and SCs association, mimicking neural tissue stiffness and reducing inflammatory responses, increase vascularization and recovery (BBB score of 8, compared to 4 in the SCI group).	[107]
	Graphene oxide	Biocompatibility and cell infiltration to enhance SC adhesion and neurite extension.	[120]
	SHIELD hydrogel (peptide-based, thixotropic, self-healing)	A 740% increase in local cell delivery, with 96% cell viability after injection compared to saline transplantation. Reduced secondary injury response and increased recovery at 4 weeks (grip strength and horizontal ladder walk).	[76]
	PLGA scaffold	SC survival and alignment as transplant substrate.	[121]
	Aligned nanofiber	Act as guiding scaffold design, and direct SC alignment.	[28]
	PCL nanofiber substrate with surface topography	2-fold increase in number of oriented neurites with nanofibers and axon elongation up to 1750 μm to the interface zone. Reduces astrocyte reactivity.	[122]
	GelMA hydrogel with activated SCs	An 85% cell viability, increased cell differentiation and growth factors expression, increased average axon length (from 100% in SCI to 160% in GelMA) improved motor recovery (BBB of 16 at 6 weeks versus 8 in controls), reduced lesion size via p38 MAPK inhibition.	[75,102,123,124,125]
Hybrid	PuraMatrix peptide hydrogel	A 25% increase in SC survival in vitro, enhanced motor recovery (BBB score of 13.3 in the Puramatrix group at 8 weeks, versus 4.7 in the lesion only).	[126]
	Matrigel And Injectable ECM hydrogels	Supportive environment for SC graft retention, reduced cavity formation, host tissue integration. Improved host–graft interface with 10-fold increase in GFAP^+^ processes in SC bridge. SC survival rates up to 36% in Matrigel at 12 weeks, with 130% increase in vascularization and 97% increase axonal in-growth compared to suspension, and increase recovery, BBB score of 11.1 at 9 weeks in SC-Matrigel implant versus 9.8 in SC-medium controls.	[25,55,108]
	Laminin-coated guidance channels	Reduced lesion cavities, glial reaction and inflammation. Oriented axonal growth when filled with SCs and neurotrophic factors.	[127]
	FGF-loaded hydrogel	Promotes remyelination and axon regeneration, increased BBB score up to 15 at 28 days, versus 5 in SCI.	[128]
	Self-healing hydrogel with neurotrophic factor	Up to 84% MBP^+^ bridge area, 1000-fold increased gene delivery. 92% of channels penetrated by axons when cells filled versus 58% when non cell filled. Increase MEP amplitude and BBB score (6 at 8 weeks, compared to 2 in the control group).	[91,114,129,130]

## 4. Translation of Combined SC and Biomaterial Approaches to Human SCI

While the combination of biomaterials with SCs holds strong potential for enhancing outcomes in SCI, several challenges remain before these approaches can be effectively translated to clinical settings. Key concerns include ensuring the long-term survival and integration of transplanted cells, optimizing scaffold properties, and addressing the feasibility of advancing such therapies into clinical trials. Despite ongoing challenges, the clinical translation of SC-based therapies is progressing, with Phase I/II trials providing foundational insights into safety, feasibility, and technical parameters for autologous SC transplantation in human SCI [137] (Table 3). Early Phase I trials have demonstrated the safety and feasibility of SC delivery in patients [23,24,138,139,140] with acute, subacute and chronic, thoracic and cervical, complete and incomplete SCI [23,24,141], while also addressing critical aspects such as SC harvesting, identification, and purification from peripheral nerves. Through long-term follow-up, up to five years after transplantation, these studies have defined and characterized delivery techniques, including the use of hypodermic 4G needles, Hamilton syringes, or tubing inserted through a pial opening, as well as dosing parameters (ranging from 50 µL containing 1 million cells to a maximum volume of 750 µL with 75 million cells) [23,24,141,142,143]. Building on this work, more recent Phase II investigations have explored combinatorial strategies, such as the intrathecal co-administration of SCs with MSCs or OECs, showing promising preliminary effects on neuropathic pain and bladder function [144,145,146]. Collectively, these clinical efforts not only validate the safety of SC-based approaches but also underscore the therapeutic potential of combining SCs with other cell types to enhance outcomes in SCI patients. Complementing clinical advances, preclinical research has demonstrated that the use of SCs with biomaterials, especially scaffolds engineered with specific topographical and biochemical cues, can significantly enhance neural regeneration. Scaffold design is crucial for guiding axonal growth, supporting transplanted cell survival, and modulating the injury microenvironment, thereby addressing key obstacles to effective repair. A persistent challenge that remains is the failure of regenerating axons to extend beyond the caudal graft-host interface. In many models, axonal growth stalls at the caudal scaffold border, failing to establish functional connections. This is often due to glial scarring, inhibitory ECM, the chemoattractant nature of the graft region compared to the caudal cord and mismatched neurotrophic gradients. Moreover, while histological improvements are frequently reported, corresponding functional gains tend to be modest or delayed, with BBB scores exhibiting improvements of several points by 8 weeks post lesion (see Table 2). This highlights the disconnect between structural repair and meaningful neurological recovery, and shapes the need to address other readouts, such as electrophysiological properties, to target connectivity and transmission of neural information. Translation to chronic settings also demands further innovation, such as scar resection or the incorporation of pharmacological modulators, to enhance receptivity to regenerative interventions. Looking ahead, several novel biomaterials used in CNS repair have yet to be explored in combination with SCs. These include zwitterionic hydrogels [147] for reduced immune reactivity, conductive polymers like polypyrrole [148] for neuromodulation, and two-dimensional nanomaterials such as MXene [149], which offer electroconductive and antioxidant properties. Their physical and biochemical characteristics make them attractive candidates for next-generation scaffolds.

However, despite these promising preclinical results, the clinical viability of these different combinatory approaches of SCs with biomaterials remains uncertain. Therapeutic manufacturing, regulatory hurdles, inter-patient variability, and the scalability of cell and biomaterial production pose significant barriers. In particular, the complexity and cost associated with personalized autologous cell therapies may limit widespread adoption unless off-the-shelf allogeneic solutions or standardized protocols can be developed. A long-standing goal in SCI research is to develop therapeutic strategies that are effective and translatable to clinical applications. Before advancing to clinical trials, experimental interventions must demonstrate efficacy and reproducibility in large animal models, yet significant differences between preclinical studies and clinical realities persist [150,151]. Anatomical and physiological differences limit the translational relevance. Rodents have smaller spinal cords with different tract sizes, locations and functions, notably a proportionally smaller corticospinal tract, which impairs modeling of long-distance axonal regeneration essential for human recovery. Additionally, recovery timelines differ markedly, weeks in rodents compared to months in humans, complicating therapy extrapolation. SCI itself is heterogeneous in anatomy and severity. Complete injuries require bridging non-neural lesion cores, while incomplete injuries retain some neural pathways. Partial lesion models help study regeneration but do not capture human SCI complexity. Timing is also critical: acute treatments focus on neuroprotection but face ethical and regulatory delays, as an invasive clinical intervention usually occurs 2–3 weeks post-injury. Chronic injuries, representing most clinical cases, involve scar formation, inflammation, and disrupted circuits, further hindering regeneration. Therefore, therapeutic windows must be adapted to an injury stage for optimal outcomes.

Combinatorial strategies, particularly those integrating SCs with biomaterials, have shown promise in preclinical models. Nevertheless, translating these findings into effective clinical therapies involves addressing numerous challenges related to cell sourcing, delivery, biomaterial standardization, and integration with the host tissue. Cell production and sourcing remain major concerns [152]. Autologous SCs require invasive nerve biopsies and may yield insufficient cell numbers, especially in elderly or severely injured patients. Expansion to therapeutic doses under Good Manufacturing Practice (GMP) conditions is also time-consuming and costly. Alternative sources, such as SCs derived from stem cells or iPSCs, offer scalability but introduce safety concerns, including tumorigenicity and variable differentiation potential. Allogeneic SCs are scalable but raise immunogenicity and survival issues post-transplantation. Biologically, SCs often fail to migrate beyond the graft core or integrate functionally, particularly in chronic lesions marked by robust glial scarring and inhibitory ECM. Clinical implementation further depends on optimizing delivery routes and therapeutic timing. Intralesional injections, while effective in rodents, may face lesion variability and accessibility in patients. Given the logistical and biological complexities of cell-based therapies, alternatives such as SC-derived products and particularly small extracellular vesicles (sEVs) are being explored [153,154,155]. These nanoscale, lipid bilayer-enclosed particles, secreted by all cell types, are key mediators of intercellular communication. sEVs retain many of the therapeutic properties of their parent cells [156], including immunomodulation and trophic support, while offering lower immunogenicity and improved stability, storage, and delivery. Their cell-specific signaling, stability, and ability to reflect the physiological state of the SCs from which they are derived make them promising candidates for incorporation into biomaterials or for systemic delivery, potentially overcoming biological complexity and compatibility of cell transplantation.

In parallel, biomaterial translation faces its own set of hurdles. While many scaffolds have demonstrated efficacy in animal models, few meet the strict standards required for human clinical use. Regulatory agencies demand precise characterization of material properties, degradation kinetics, and reproducibility. Biocompatibility, sterilization, and storage stability must also be validated. Moreover, the biomechanical properties of biomaterials, including stiffness and degradation rate, must be tuned to support both SC viability and host tissue remodeling over time. Beyond biosafety, lot-to-lot consistency, and ensuring the biomaterial or hydrogel can be injected through a 25-G needle if an injectable gel [110,152], or the need for spinal cord debridement when a solid 3D scaffold is implanted, other requirements are imposed by the FDA including: (1) cGMP-compliant production processes; (2) sterility and endotoxin assurance; (3) validated assays for identity, potency, and shelf-life stability; (4) mechanical integrity testing of the scaffold; (5) viral and adventitious-agent safety for any biologic component; and (6) backed by GLP pre-clinical data before first-in-human use.

Additionally, synergistic combinations with other therapeutic modalities, such as neuromodulation, rehabilitation, and pharmacological agents, could be necessary to address the multifactorial nature of SCI obstacles to repair. Beyond stimulating endogenous axon regrowth or establishing new connections through cell grafts, the presence and transmission of neural information alone are not sufficient to induce robust volitional motor behavior or circuit reorganization, which requires activity-dependent mechanisms [157]. Accordingly, emerging neurotechnological strategies designed to enhance neuroplasticity and functional restoration, including electrical stimulation and brain–computer interfaces [158,159,160], along with activity-based therapies, have emerged as a key approaches. By promoting recovery through repeated activation of the sensorimotor system that enhances activity-dependent plasticity in both spared and newly formed circuits, these therapies may ultimately lead to functional improvements [157,161].

While experimental advances have improved our understanding of the pathophysiology of SCI and the reorganization of neuronal circuits, significant anatomical and physiological differences exist between rodents and humans. These differences include the size and distribution of ascending and descending tracts, the extent of gray matter to reinnervate, and the timeline of recovery, which is much slower in humans. Additionally, human SCI lesions are highly heterogeneous across individuals, varying in location and severity, and are neither homogenous nor reproducible, due to the existence of different morphologies (complete vs. incomplete), that vary in size and organization [5]. This poses major challenges for translational relevance. While anatomically complete lesions require bridging large non-neural lesion cores, incomplete injuries may retain enough neural tissue to support spontaneous communication. Moreover, treatment efficacy depends heavily on timing post-injury, as therapeutic responses differ across acute and chronic phases. Early interventions may offer neuroprotection but can be challenging to translate in a clinical trial setting, where informed consent typically delays treatment by 2–3 weeks [16,151]. Such delays may contribute to the discrepancies between animal and human outcomes. Although lesion progression is increasingly understood, the optimal repair timeline, especially for combinatorial approaches, remains unclear. Expanding studies into higher-order models such as non-human primates is thus critical. Finally, despite encouraging preclinical results, clinical relevance remains uncertain, as targeting all aspects of this complex pathology simultaneously may not necessarily yield additive or synergistic effects [162], emphasizing the need for rational, evidence-based therapeutic design.

**Table 3 ijms-26-07922-t003:** SC implantation for human SCI trials.

Trial	Injury	Dose	Outcomes	Refs
Clinical Phase I trial, using SC transplantation [2017; USA]	6 subjects T1-T6 SCI within 60 days post-injury	Dose escalation from 5 (50 μL) to 15 (150 μL) million cells	Increased FIM and SCIM III scores and electrophysiological activity, no motor changes, minor sensory changes (ISNCSCI and AIS). Neurological function changes after transplant not clinically meaningful in 5/6 subjects. One AIS A-to-B conversion. Lesion length and volume were significantly different between baseline and 12 mos. post-transplantation. Improved global impression of change; no severe adverse events.	[23]
Clinical phase 1 trial, using SC transplantation [2022; USA]	8 subjects C5-T11 SCI within 1 to 15 years post-injury	Dose escalation from 20 (200 μL) to 75 (750 μL) million cells	No AIS change; no significant motor improvement; improved total sensory scores (ISNCSCI); neurophysiological changes at 12 mos., including emergence of MEP and voluntary EMG activation in legs. Cardiorespiratory improvement and strength gains in some patients at 6 mos. Reduced cavity volume on 6/8 participants, by MRI imaging. No severe adverse events.	[24]
Clinical study, using SC transplantation [2012; China]	6 subjects C5-T12 SCI, within 1 week to 20 months post-injury	~5 million cells (200 μL)	Improved motor and sensory function (increased ASIA scale and FIM scores; increase in latency and amplitude of SSEPs and MEPs); improved autonomic function and quality of life. No immunological issues; no severe adverse events.	[141]
Clinical study using SC transplantation [2008; Iran]	33 subjects T6-T9 and cervical SCI, within average of 4.1 years post-injury	3–4.5 million cells (300 μL)	Improved light touch sensory scores, minimal improvement in pin-prick sensation and motor scores (ASIA). Improved bladder sensation and urinary control; no significant increase in FIM scores. Follow-up MRI showed no change in cord size, pathology, or injection-related adverse effects.	[142,143]
Clinical Phase II Trial, using combination of SCs and BMSCs [2025; Iran]	37 subjects, Thoracic and cervical SCI, 14 days to 6 months post lesion	6 mL mixture containing: MSCs: 5 × 10^5^ cells/mL SCs: 5 × 10^5^ cells/mL	Reduced neuropathic pain (ISCIPBDS and NRS scores) and increased sensory function (light touch, pinprick); improvements in general perception, health, physical, psychological, social, and environmental domains (WHOQOL-BREF). Increased urodynamic function; reduced urinary incontinence episodes; no significant improvement in bladder capacity and compliance; improved scores on I-QOL.	[144,145]
Clinical Trial, using combination of SCs and OECs [2014; China]	1 subject, C5-C7 SCI, 5 years post-injury	1 million cells (50 μL; 50% SCs, 50% OECs)	No change in neurological function (motor, light touch, pinprick and IANR scores); stronger arm movements. Increased bilateral forearm EMG recruitment.	[146]

## 5. Conclusions

In conclusion, the combination of SCs and biomaterials offers a compelling strategy that yields additive or synergistic effects that address both cellular and structural barriers to regeneration. SCs provide trophic support, remyelination, and guidance to regenerating axons, while biomaterials enhance their survival, localization, and integration through physical scaffolding, controlled release of bioactive agents, and modulation of the lesion environment. Preclinical studies consistently demonstrate superior outcomes with combinatorial approaches compared to monotherapies. Yet despite this preclinical validation, translation to human trials remains limited, constrained by challenges in cell sourcing, biomaterial standardization, and delivery logistics. To date, no clinical studies have fully implemented scaffold-supported SC transplantation in SCI patients.

However, foundational work has laid the groundwork. Several clinical trials in the US involve SCs alone, and much ongoing research now focuses on the development of biomaterial-based delivery platforms in preparation for future combination trials. Parallel efforts by research groups in Europe and Asia are optimizing injectable hydrogels and photo-crosslinkable matrices for clinical compatibility with human SCs, including iPSC-derived SC populations. Moving forward, the integration of combinatorial therapies will involve modular platforms that combine biomaterials with SCs and additional components. These multi-modal approaches offer the flexibility needed to adapt to the heterogeneity of human SCI.

Progress will depend on iterative refinements in material design, scalable cell production, and alignment with clinical trial frameworks. With advancing biomaterials technology and increasing regulatory clarity, and supported by collaborative efforts among researchers, clinicians, bioengineers, and regulatory agencies, alongside the engagement of patients, early-phase clinical trials combining SCs with biocompatible scaffolds are anticipated within the next decade.

## Figures and Tables

**Figure 1 ijms-26-07922-f001:**
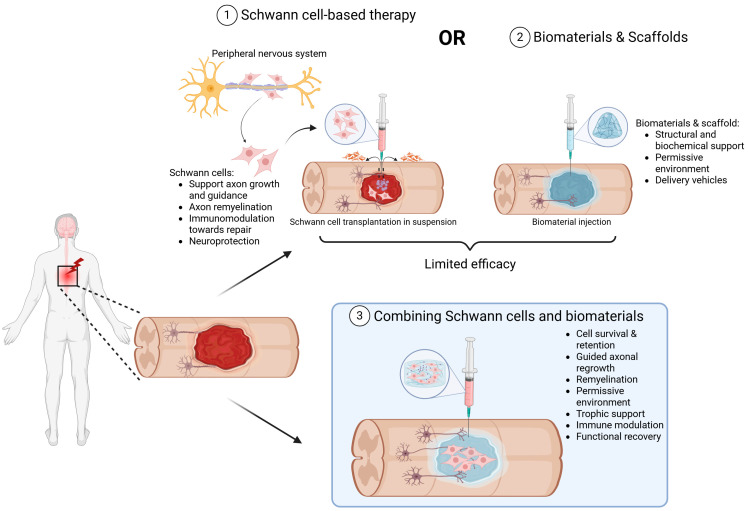
Schematic showing the advantages and limitations of SCs or biomaterials as well as the reasoning for their combinatory therapeutic use for neuroprotection and repair. (1) SCs, derived from the peripheral nervous system or from stem cells (pink), support tissue protection, repair, axonal growth, and remyelination when transplanted into SCI lesions. Transplanted SCs, however, are exposed to the hostile milieu of the injury site, including cytotoxic molecules, oxidative and hypoxic stress and cell-mediated immunity to reduce their survival (purple) and limit efficacy. (2) Biomaterials (blue) can provide structural support and protection to cells as well as deliver therapeutic molecules that can change the injury environment to one permissive for axon growth and neurorepair. (3) Combining SCs (pink) with biomaterials (blue) represents a promising strategy to enhance transplanted cell survival and persistence, stimulate axonal regeneration, provide trophic support, modulate the immune response and ultimately further promote functional recovery.

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
