# Peer review of "Engineered Healing: Synergistic Use of Schwann Cells and Biomaterials for Spinal Cord Regeneration"

_ijms, 2025, doi:10.3390/ijms26167922_

Round 1
Reviewer 1 Report
Comments and Suggestions for Authors
This manuscript describes a review of the literature on “ Engineered Healing: Synergistic Use of Schwann Cells and Biomaterials for Spinal Cord Regeneration”. Authors have summarized the recent progress and ongoing efforts in Schwann cells and biomaterial-based regenerative strategies. Schwann cells have shown potential for promoting nerve regeneration, remyelination, and neuroprotection, with clinical trials confirming their safe use in humans. However, there are several challenges face Schwann cells therapies such as limited survival, integration, and migration after transplantation. This summarized review focuses on current progress in Schwann cells -based therapies combined with biomaterials, highlighted key preclinical advances, clinical translation efforts, and the path forward toward effective regenerative interventions for SCI. The mechanism are sufficiently described by the figure and table in the manuscript. Therefore, I would like to suggest this manuscript could be accepted with the present form.
Author Response
Reviewer 1
Reviewer Comment 1.1 This manuscript describes a review of the literature on “ Engineered Healing: Synergistic Use of Schwann Cells and Biomaterials for Spinal Cord Regeneration”. Authors have summarized the recent progress and ongoing efforts in Schwann cells and biomaterial-based regenerative strategies. Schwann cells have shown potential for promoting nerve regeneration, remyelination, and neuroprotection, with clinical trials confirming their safe use in humans. However, there are several challenges face Schwann cells therapies such as limited survival, integration, and migration after transplantation. This summarized review focuses on current progress in Schwann cells -based therapies combined with biomaterials, highlighted key preclinical advances, clinical translation efforts, and the path forward toward effective regenerative interventions for SCI. The mechanism are sufficiently described by the figure and table in the manuscript. Therefore, I would like to suggest this manuscript could be accepted with the present form.
Response: We thank the reviewer for the time in evaluating our manuscript and appreciate their positive comments.
Reviewer 2 Report
Comments and Suggestions for Authors
please see uploaded file

Author Response
Reviewer 2
Reviewer Comment 2.1 The review only appears to be interested in presenting information related to combining Schwann cells and biomaterials in the context of tissue repair but completely ignores any reference to the ground-breaking works that demonstrated the feasibility of Schwann cells in their native, nerve environment to support long distance CNS axonal regeneration (David and Aguayo, 1981, and many others including Cheng et al., 1996). It seems obvious that the current technological advances of combining highly enriched populations of Schwann cells with a wide range of hydrogel and 3D scaffolds is an attempt to engineer and improve on the powerful axon growth-promoting and orientational guidance cues of implanted peripheral nerves. I would suggest that the incorporation of this important background information, should go so far as the inclusion of the grafting of multiple peripheral nerves into human spinal cord injury as part of a combinational intervention strategy involving OEC transplantation and intensive physiotherapy (Tabakow et al., 2013, 2014). Although the hype related to the single patient case reports of 2013 and 2014 drew scientific criticism for being “over-sold” in the media, and by no means established a verifiable “cure for spinal cord injury”, it did demonstrate a proof of principle that should not be ignored.
Response: While we tried to keep the review highly focused on the specific topic of combined SCs and biomaterials, and particularly advances made in the past decade, we understand that some further elaboration on the seminal work which spurned this research endeavor would provide a stronger foundation for the review. We have now expanded Section 2 to include the important work that the reviewer has identified (See lines 147-164)
Reviewer Comment 2.2 Lines 260-263: “Luo et al.[80] demonstrated that microgrooved chitosan scaffolds seeded with SCs improved cell orientation, upregulated neurotrophic gene expression, enhanced remyelination and motor recovery in a rat SCI model. “ There seems to be an error here – the Luo manuscript refers to hydrogel investigations and its citation here is wrong.
Response: Thank you for pointing this typo out. The manuscript to be cited here was the work of Cong et al. 2024, not Luo. We also realized this paragraph, that comprised several different types of discussions, was confusing. We have therefore corrected the references and improved the clarity of the paragraph (See lines 327-330)
Reviewer Comment 2.3 Line 355: this statement is missing a citation – the Luo citation.
Response: Thank you for pointing this out. In response we have included the appropriate citation.
Reviewer Comment 2.4 Intro line 40: The authors describe a WHO statement “prevalence of SCI at 250 000 and 500 000 per year worldwide ...”. I think that there is a misunderstanding here. This is more likely to be incidence (numbers relating to 2019), rather than prevalence, which worldwide is in the millions. Please double check.
Response: Thank you for pointing this out. We have corrected this, updating the numbers with more recent data and appropriate citations. (See lines 42-48)
Reviewer Comment 2.5 Line 95: “(including exoskeletons, functional electrical stimulation, brain–computer interfaces)“ - although the authors rightly mention the development of alternatives to tissue engineering and regenerative medicine, the examples of exoskeleton and brain-computer interfaces lack any appropriate accompanying citation. This should be addressed.
Response: Thank you for the suggestion. While this section was initially considered to be potentially outside the focus of the review, we agree with the reviewer that it represents an important area for alternative and combinatory approaches. We have therefore decided to retain it and have included appropriate citations from the relevant field to support its inclusion. (See lines 563-574)
Reviewer Comment 2.6 Section 2 title, line 125: “adaption” or “adoption”?
Schwannosis after human spinal cord injury has been known for several decades. Since the current submission describes the development of Schwann cell/biomaterial based strategies for treating human spinal cord injury, the authors’ decision to cite the relatively recent publication of Miranda et al., (2019, #36) which describes nontrauma-related Schwannosis in foals and a calf seems difficult to understand. I would suggest that Bruce et al., (2000) J. Neurotrauma 17: 781-788 would be more relevant.
Response: We thank the reviewer for this comment. We have corrected the title to “adaption”, added the more relevant citation and introduced an explanation of the term to improve clarity. (See lines 124 and 162)
Reviewer Comment 2.7 Lines 206-209: “ Biomaterials can be tailored in terms of stiffness, degradation kinetics, and bioactivity to match the regenerative needs of spinal cord tissue but also in term of porosity, permeability, mesh size, mechanical and rheological properties…”. This information would be greatly improved by providing indicating which rheological value ranges have been found to be appropriate for materials intended to be implanted into spinal cord. Also, which degredation characteristics are regarded as being optimal. Include appropriate citations please.
Response: Thank you for this insightful suggestion and adding this specific information would indeed be helpful to the reader. In response, we have expanded the Discussion section to more explicitly address the rheological properties and ideal degradation characteristics of biomaterials, in general and in the context of SCI implantation, supported by appropriate references. (See lines 229-247)
Reviewer Comment 2.8 Lines 210-212 “They can be formulated as injectable gels or preformed scaffolds. Finally, their structural features, such as channels, nanotopography, or capillary architecture, can support directional axonal regrowth, vascularization, and integration.” This statement covers an enormous amount of research but is not accompanied by any appropriate citations – this omission should be corrected.
Response: We have included appropriate citations for each of these advancements as requested. (See line 250)
Reviewer Comment 2.9 Lines 215-216: “ Moreover, inflammation and mechanical mismatch at the implant site may limit their long-term function.” Again, these statements need to be backed up with appropriate citations.
Response: We have revised the paragraph to include appropriate citations related to the integration of materials after implantation, as well as general properties of biomaterials, such as their potential to induce inflammation, and, more broadly, their advantages/disadvantages, as suggested in comment 2.10. (See lines 248-274)
Reviewer Comment 2.10 Lines 221-223: “These materials offer structural support, protect cells from mechanical stress during injections, and create a permissive microenvironment for survival, integration, and axonal regrowth.” This is a valid comment about the biomaterials used as hydrogels or 3D scaffolds, but the authors should also mention the relative advantages/disadvantages of hydrogels and 3D scaffolds when it comes to applying them to the lesion site.
Response: We agree with the reviewer on the importance of including a critical discussion. In response, we have outlined the advantages and disadvantages of these materials, specifically in the context of transplantation for SCI, and have supported this with appropriate citations. (See lines 248-274)
Reviewer Comment 2.11 Lines 226-229: “For example, gelatin hydrogels[68] in rat transection models, alginate-based scaffolds combined with SC transplantation and neurotrophic factor delivery[69–71], and fibrin matrices ([65], in hemisection models) have shown promising outcomes”. The authors need to provide more detail about what they mean by “promising outcomes”.
Response: We now include specific details on the promising outcomes reported in these studies. (See lines 278-287)
Reviewer Comment 2.12 Lines 229-230: “In vitro studies have also highlighted the benefit of combining SCs with fibrin[72] and hyaluronic acid (HA)-laminin composites[73].“ What are these highlighted benefits? More detail required.
Response: We now include in the revised manuscript more detailed information regarding the highlighted benefits, specifically related to cell viability and the guidance of regenerating axons. (See lines 287-292)
Reviewer Comment 2.13 Lines 238-240: “Study have reported up to 75% survival of rubrospinal neurons and an increase in axon numbers depending on scaffold composition (17-fold for alginate-trophic factor combinations, versus 6–7-fold for PEG-based scaffolds)[74].” More detail required please.
Response: We have included additional details about the exact conditions under which these results were observed, including cell type, biomaterial type, controls, delivery method, and injury paradigm. (See lines 293-311)
Reviewer Comment 2.14 Lines 280-283: “Similarly, Chen et al.[85] used GDNF-secreting SCs delivered via OPF. or PEG hydrogels in a thoracic transection model, resulting in enhanced axon growth, remyelination, and motor improvement within three weeks post-injury.” The presentation of this data doesn’t include a possible interpretation of the rapid, but modest motor improvement over the short, post-operative period being due to the effect of released GDNF on spinal tissue caudal to the implant. Wouldn’t this addition be more “balanced”?
Response: Thank you for the suggestion. We agree with the reviewer on the need to include constructive and balanced criticism. We have therefore incorporated possible interpretations attributing effects to factors other than the hydrogel-cell construct itself, including the timing and duration of growth factor release, considering the research paradigm. (See lines 350-354)
Reviewer Comment 2.15 Line 291: the authors should explain what is meant by the term “shelf-healing” hydrogel.
Response: While we did not find the term “Shelf-healing” in the text, we did mention the “SHIELD” biomaterial from the referenced paper, which is the name given to this hydrogel by the authors. However, we agree with the reviewer that our discussion also included self-healing hydrogels (including SHIELD), which was insufficiently detailed. We have now added more information to better explain this properly. (See lines 362-366)
Reviewer Comment 2.16 Lines 341-343: “Injectable hydrogels derived from decellularized porcine peripheral nerve tissue have also shown high translational potential.“ The authors should explain why hydrogels from porcine sources would show a high translational potential. It was mentioned that Matrigel being derived from murine sarcoma was part of the reason that restricted its clinical translation. Please add a note of explanation as to how the porcine source of the hydrogel doesn’t similarly limit its translational potential.
Response: Thank you for pointing this out. While hydrogels derived from decellularized porcine peripheral nerve tissue originate from animals, which could limit their translational potential, the preparation process includes decellularization and neutralization steps designed to remove all cellular components and significantly attenuate immunogenicity. There is also the potential to employ humanized animals with this approach. We have incorporated these clarifications in the corresponding section. (See lines 423-428)
Reviewer 3 Report
Comments and Suggestions for Authors
Despite extensive research, a universally accepted regenerative therapy for spinal cord injuries remains elusive. However, biomaterial therapy emerges as a promising technique among various strategies aimed at treating these injuries. Presently, a combination of multiple cell therapies and biomaterials is being explored to synergistically promote the regeneration of damaged spinal cord tissue. This manuscript provides a comprehensive review of the research progress on the utilization of Schwann cells and biomaterials in treating spinal cord injuries, with a particular focus on the preliminary effects observed in human implantation therapy. The review aligns with the scope of this journal and is structured in a clear, focused, and well-written manner, offering insightful information on biomaterial therapy for spinal cord injuries. I recommend removing the content about peripheral nerves in SECTION 2 of the manuscript.
Author Response
Reviewer 3
Reviewer Comment 3.1 Despite extensive research, a universally accepted regenerative therapy for spinal cord injuries remains elusive. However, biomaterial therapy emerges as a promising technique among various strategies aimed at treating these injuries. Presently, a combination of multiple cell therapies and biomaterials is being explored to synergistically promote the regeneration of damaged spinal cord tissue. This manuscript provides a comprehensive review of the research progress on the utilization of Schwann cells and biomaterials in treating spinal cord injuries, with a particular focus on the preliminary effects observed in human implantation therapy. The review aligns with the scope of this journal and is structured in a clear, focused, and well-written manner, offering insightful information on biomaterial therapy for spinal cord injuries. I recommend removing the content about peripheral nerves in SECTION 2 of the manuscript.
Response: We thank the reviewer for the time in evaluating our manuscript and appreciate both the positive comments and constructive feedback. Regarding the suggestion to remove content related to peripheral nerves in Section 2, this part was included to provide important context and a fundamental background for the use of SCs for SCI applications. As other reviewers recommended keeping or expanding this section, particularly to highlight seminal studies that provided the initial support for using SCs in CNS repair, we have decided to keep this section but narrow the focus specifically to those topical areas that other reviewers identified as important for contextual purposes.
Reviewer 4 Report
Comments and Suggestions for Authors
This paper is a well-considered and comprehensive review that successfully integrates current developments in the combined use of Schwann cells (SCs) and biomaterials for the repair of spinal cord injury (SCI). The manuscript presents a logical flow that extends from the pathophysiology of SCI through to the challenges of translation, while providing a critical evaluation of preclinical studies and clinical trials. It is particularly commendable for its balanced integration of both biological and engineering perspectives.
The article is nicely organized and replete with citations, figures, and comparison tables. Nevertheless, there are a number of places that would require more specific articulation, greater critical analysis (particularly in translational limitations), and better clarity in certain figure legends and nomenclature.
1. The abstract is informative but would be improved by more clarity on the *unique advantages* of pairing SCs with biomaterials, particularly in relation to other cell types (e.g., NSCs, MSCs).
2. The legend to the figure is not detailed enough. Please indicate what each part (SCs, types of scaffolds, lesion areas) in the scheme depicts. Mention legend color codes and abbreviations.
3. Lines 36–68 can be summarized. The amount of detail on lesion compartments is perhaps too much for a general introduction and can be transferred to subsequent sections if necessary.
4. Although Tables 1 and 2 are useful, some of the statements are qualitative. Wherever possible, include more direct data (e.g., % axon growth, BBB score improvements) for side-by-side comparison of biomaterials.
5. The limitations discussion (lines ~366–411) is a bit descriptive. A more critical commentary regarding the *clinical viability* and *realistic timelines* for translation would make the conclusion stronger.
6. The review emphasizes histological and behavioral outcomes but largely omits electrophysiological readouts, which are critical for functional integration.
7. The conversation should elaborate on manufacturing challenges, FDA regulatory hurdles, and patient variability, with particular reference to chronic SCI.
8. Are there any studies directly comparing SCs + biomaterials with NSCs or OECs + biomaterials? The inclusion of such comparisons would place SCs' relative effectiveness in perspective.
9. Generally strong, but some excessively long or complicated sentences (e.g., lines 90–97 and 365–377). Try breaking them up for better readability.
10. Pearse et al. is cited frequently in the manuscript, and while this is acceptable, caution must be exercised not to come across as excessively self-referential. Wider contextualization would add objectivity.
11. Inconsistency in the use of acronyms (e.g., "SCs" vs. "Schwann Cells") and formatting of references [e.g., some references include full titles, others do not].
12. Table of major Phase I/II trials with SCs (dosing, type of injury, outcomes) would be a great addition to the "Translation" section.
13. The conclusion mention of MXene, polypyrrole, and zwitterionic gels is great. Try to expand this in the main text instead of just mentioning them briefly under future directions.
14. Terminology such as "schwannosis" and "Bands of Büngner" is used with minimal explanation. Define briefly on first use for general audience understanding.
15. EVs are touched upon towards the end, but further discussion of their research status, preclinical results, and delivery issues would be enlightening.
Comments on the Quality of English LanguageThe quality of English language in this manuscript is generally high and professionally written. The authors demonstrate a strong command of scientific terminology and effective structure for a review article. However, there are a few areas where the clarity and readability can be improved.
Author Response
Reviewer 4
Reviewer Comment 4.1 The abstract is informative but would be improved by more clarity on the *unique advantages* of pairing SCs with biomaterials, particularly in relation to other cell types (e.g., NSCs, MSCs).
Response: We agree with the reviewer that the unique advantages of Schwann cells were presented somewhat vaguely. As suggested, we have clarified this point, specifically in relation to other cell types. (See lines 25-28)
Reviewer Comment 4.2. The legend to the figure is not detailed enough. Please indicate what each part (SCs, types of scaffolds, lesion areas) in the scheme depicts. Mention legend color codes and abbreviations.
Response: We have updated the figure legend to include more details and the color codes as requested. (See lines 112-122).
Reviewer Comment 4.3. Lines 36–68 can be summarized. The amount of detail on lesion compartments is perhaps too much for a general introduction and can be transferred to subsequent sections if necessary.
Response: We agree with the reviewer that this section appeared overly expanded. In response, we have summarized the Introduction to better focus on the review’s objective. (See lines 49-61)
Reviewer Comment 4.4. Although Tables 1 and 2 are useful, some of the statements are qualitative. Wherever possible, include more direct data (e.g., % axon growth, BBB score improvements) for side-by-side comparison of biomaterials.
Response: As suggested, we have included more direct and quantitative data from the reference papers where available. (See line 455)
Reviewer Comment 4.5. The limitations discussion (lines ~366–411) is a bit descriptive. A more critical commentary regarding the *clinical viability* and *realistic timelines* for translation would make the conclusion stronger.
Response: We have addressed this issue by adding a more critical discussion focused on human translation, particularly regarding clinical availability and the chronic timeline. These aspects are also aligned with the changes we made to address regulatory hurdles, as noted in comment 4.7. (See lines 488-523)
Reviewer Comment 4.6. The review emphasizes histological and behavioral outcomes but largely omits electrophysiological readouts, which are critical for functional integration.
Response: Thank you for this valuable insight. We have now incorporated the discussion of these outcomes with more emphasis for those reports that employed such measures In addition, in the discussion we now discuss potential biomaterials that may be more amenable to strategies employing neuromodulation in future directions, for further improving outcomes as suggested by reviewer’s comment 4.13. (See lines 407-412; 438-449)
Reviewer Comment 4.7. The conversation should elaborate on manufacturing challenges, FDA regulatory hurdles, and patient variability, with particular reference to chronic SCI.
Response: While these aspects were briefly introduced earlier in the text, we agree with the reviewer that they were only mentioned briefly. We have now revised the discussion to include more detailed information and appropriate references addressing FDA hurdles as well as patient variability in lesion characteristics and time course. (See lines 549-562)
Reviewer Comment 4.8. Are there any studies directly comparing SCs + biomaterials with NSCs or OECs + biomaterials? The inclusion of such comparisons would place SCs' relative effectiveness in perspective.
Response: To our knowledge, there is a dearth of such studies, though several published works have compared different cell types in SCI models without biomaterials. We have identified a study by Olsen and colleagues that did compare, in a complete SCI paradigm, SCs and NSCs in biomaterials and that work is now discussed in Section 2 of the revised manuscript (See lines 212-220)
Reviewer Comment 4.9. Generally strong, but some excessively long or complicated sentences (e.g., lines 90–97 and 365–377). Try breaking them up for better readability.
Response: We have broken up long or complicated sentences as requested in the lines mentioned as well as elsewhere in the revised manuscript to ensure clarity and improve readability.
Reviewer Comment 4.10. Pearse et al. is cited frequently in the manuscript, and while this is acceptable, caution must be exercised not to come across as excessively self-referential. Wider contextualization would add objectivity.
Response: We understand the reviewer’s critique. In addition to the numerous contributions to this area of research made by our group we now provide additional references that validated, built upon or reported other related findings in these specific sections to provide expanded discussion as well as wider contextualization and objectivity as suggested.
Reviewer Comment 4.11. Inconsistency in the use of acronyms (e.g., "SCs" vs. "Schwann Cells") and formatting of references [e.g., some references include full titles, others do not].
Response: We have reviewed the acronyms and references and have corrected the text accordingly.
Reviewer Comment 4.12. Table of major Phase I/II trials with SCs (dosing, type of injury, outcomes) would be a great addition to the "Translation" section.
Response: We thank the reviewer for this comment and agree that Phase I/II studies represent a major step toward the clinical use of these cells, especially when focusing on the next steps involving their putative combination with biomaterials. To ensure alignment with the selected focus of the review, we have chosen to include these outcomes with appropriate references in the translation section rather than creating a dedicated table. (See lines 462-476)
Reviewer Comment 4.13. The conclusion mention of MXene, polypyrrole, and zwitterionic gels is great. Try to expand this in the main text instead of just mentioning them briefly under future directions.
Response: Thank you for the suggestion. These materials were initially introduced as future directions, considering that they have not yet been investigated in combination with SCs. However, as also pointed out by the reviewer in comment 4.6, we agree on the importance of including electrophysiological readouts. Therefore, we have expanded the discussion of these materials to expand this aspect, rather than briefly mentioning them at the end of the review. (See lines 490-500)
Reviewer Comment 4.14. Terminology such as "schwannosis" and "Bands of Büngner" is used with minimal explanation. Define briefly on first use for general audience understanding.
Response: We agree with the reviewer, as well as others who pointed this out, that specific terms require explanation for better clarity. We have now included explanations along with relevant references. (See lines 134 and 162)
Reviewer Comment 4.15. EVs are touched upon towards the end, but further discussion of their research status, preclinical results, and delivery issues would be enlightening.
Response: Thank you for the suggestion. We have therefore updated the discussion to include expanded information on the use of EVs as “cell-derived products,” while maintaining the focus of the review on SC applications with biomaterials, by presenting them as an alternative avenue. (See lines 538-548)
Round 2
Reviewer 4 Report
Comments and Suggestions for Authors
The authors have addressed reviewer feedback from the previous round very thoroughly. In most instances, the revisions have noticeably enhanced the manuscript in terms of clarity, contextualization, and scope of coverage. Overall, the manuscript is in a greatly improved form and is ready for publication subject to a few minor refinements detailed below.
Minor Comments
1. The section on the lesion compartment (Lines ~49–61) is still a bit detailed for a general introduction. Try to summarize by 2–3 sentences to maintain the focus on the review's purpose.
2. Although readability is better, there are still some multi-clause sentences (e.g., Lines ~54–61). Splitting these into two shorter sentences would enhance clarity even more.
3. Table 1 is mostly qualitative. If there are any quantitative measures (e.g., percentages of axon growth, functional scores) available for biomaterials listed, include them for the sake of consistency with Table 2.
4. The inclusion of clinical trial data in the translation work is a good decision, but a concise table of trial design, dosing, injury type, and primary outcomes would improve ease of access.
5. Though better, review early parts to make sure that after "Schwann cells (SCs)" is introduced, the abbreviation is consistently used and not sporadically substituted with the full phrase for no apparent reason.
Author Response
Comment 1. The section on the lesion compartment (Lines ~49–61) is still a bit detailed for a general introduction. Try to summarize by 2–3 sentences to maintain the focus on the review's purpose.
Response: We have rewritten this section into two shorter sentences. (see lines 49-52)
Comment 2. Although readability is better, there are still some multi-clause sentences (e.g., Lines ~54–61). Splitting these into two shorter sentences would enhance clarity even more.
Response: We have improved readability by shortening long sentences at the lines identified.
Comment 3. Table 1 is mostly qualitative. If there are any quantitative measures (e.g., percentages of axon growth, functional scores) available for biomaterials listed, include them for the sake of consistency with Table 2.
Response: We appreciate the reviewer’s suggestion. We have included all quantitative measures available in the referenced studies. However, in this specific aspect of Schwann cell–biomaterial combinations, most early investigations focus on parameters such as cell viability and integration, with limited exploration of regeneration outcomes. When regeneration is assessed, it is often reported qualitatively.
Comment 4. The inclusion of clinical trial data in the translation work is a good decision, but a concise table of trial design, dosing, injury type, and primary outcomes would improve ease of access.
Response: We thank the reviewer for this suggestion. We have now included a table summarizing the clinical trial data to facilitate access and improve readability. (see line 586)
Comment 5. Though better, review early parts to make sure that after "Schwann cells (SCs)" is introduced, the abbreviation is consistently used and not sporadically substituted with the full phrase for no apparent reason.
Response: We have reviewed the manuscript to ensure consistency in the use of the term "Schwann cells (SCs)". After the initial introduction in the abstract, the abbreviation SCs is used throughout the main text, with the full phrase retained only in the title, keywords and the abbreviation list for clarity.